SOFTWARE

# napariTFM: An open-source tool for traction force microscopy and monolayer stress microscopy

Artur Ruppel[1¤]*, Dennis Wörthmüller[2], Martial Balland[3], François Fagotto[1]

**1** Centre de Recherche en Biologie Cellulaire de Montpellier (CRBM), UMR5237, University of Montpellier and CNRS, France, **2** Institut Curie, Université PSL, Sorbonne Université, CNRS, Physique des Cellules et Cancer, Paris, France, **3** Univ. Grenoble Alpes, CNRS, LIPhy, Grenoble, France

¤ Current address: Cell Polarity, Migration and Cancer Unit, Équipe Labellisée Ligue Contre le Cancer 2023, Institut Pasteur, UMR3691 CNRS, Université Paris, Paris, France
* artur.ruppel@pasteur.fr

## Abstract

Cellular force generation and transmission are fundamental processes driving cell migration, division, tissue morphogenesis, and disease progression. Traction Force Microscopy (TFM) and Monolayer Stress Microscopy (MSM) have emerged as essential techniques for quantifying these mechanical processes, but current software solutions are fragmented across multiple platforms with varying degrees of usability and accessibility. Here, we present napariTFM, a comprehensive open-source plugin for the napari image viewer that integrates state-of-the-art algorithms for both TFM and MSM analysis within an intuitive graphical user interface. The software implements TV-L1 optical flow for displacement analysis, Fourier Transform Traction Cytometry (FTTC) for force reconstruction, and finite element methods for stress calculation, supporting both single-frame and time-series analysis of 2D microscopy data. Systematic validation using synthetic datasets with known ground truth values demonstrated excellent accuracy, with correlation coefficients above 0.9 for most situations. Real-time parameter adjustment and immediate visualization capabilities enable interactive optimization of analysis parameters and quality assessment during processing. Finally, we demonstrate the software's capabilities through analysis of optogenetic contractility experiments in cell doublets. napariTFM addresses critical gaps in the cellular mechanics software ecosystem by combining algorithmic rigor with practical usability, providing the research community with an accessible platform for quantitative studies of cellular force generation and transmission.

## Author summary

Cell-generated forces are essential in driving and regulating numerous biological processes, including development, wound healing, and disease progression.

**Data availability statement:** All data and code is available on https://github.com/ArturRuppel/napariTFM.

**Funding:** This work was supported by the ANR (Agence Nationale de la Recherche) grant Inters-cal (ANR-21-CE13-0042), coordinated by FF, which also funded AR's salary. DW received funding from a European Research Council (ERC) grant ERC-SyG 101071793, which also supports his salary. MB received funding from Programme Interne Blanc (PIB 2025). The funders had no role in study design, data collection and analysis, decision to publish, or preparation of the manuscript.

**Competing interests:** The authors have declared that no competing interests exist.

Traction Force Microscopy (TFM) is a widely used technique to measure these cell-substrate forces. Cells are cultured on elastic gels of known stiffness with embedded fluorescent beads. By comparing images of the gel in deformed (cell-attached) and relaxed states (cell-detached), bead displacements can be measured and converted into traction forces through Fourier-based mathematical methods. For cells in monolayers, Monolayer Stress Microscopy (MSM) extends this analysis to calculate internal tissue stresses, assuming the cell layer behaves as a homogeneous and linear material. Despite TFM's widespread use, existing software tools remain fragmented and often require programming expertise. napariTFM addresses this gap with a comprehensive, user-friendly platform built on napari, a python-based image viewer. The software combines an intuitive graphical interface for interactive parameter exploration with robust batch processing for large datasets, making quantitative force measurements accessible to researchers without computational backgrounds.

## 1 Introduction

Cellular force generation and transmission are fundamental processes that drive and regulate critical biological functions including cell migration, division, tissue morphogenesis, and disease progression [1–3]. Over the past decades, Traction Force Microscopy (TFM) and Monolayer Stress Microscopy (MSM) have emerged as powerful techniques for quantifying these mechanical processes, enabling researchers to measure cell-substrate forces and internal cellular stresses with high spatial and temporal resolution [4–6].

TFM reconstructs cellular traction forces by measuring substrate deformations caused by adherent cells on elastic substrates embedded with fiducial markers [7]. The technique typically involves comparing images of fluorescent beads in stressed (cell-attached) and relaxed (cell-removed) states, followed by computational reconstruction of force fields.

Several computational approaches have been developed for this inverse problem, including Boundary Element Methods (BEM) [8], Fourier Transform Traction Cytometry (FTTC) [9,10], and finite element approaches [11,12], each with distinct advantages and limitations depending on the experimental context.

MSM extends this analysis by calculating internal stress distributions within cell monolayers [6,13]. By modeling cellular tissues as thin elastic sheets, MSM enables determination of intercellular force transmission and stress propagation. Important contributions from the biomechanics community have advanced our understanding of these methods' capabilities and limitations, including developments in 4D TFM [14], quantification of active versus resistive stresses [15], and theoretical frameworks for stress inference in confluent tissues [16].

Despite the widespread adoption of these techniques, significant barriers limit their accessibility to the broader biological research community. Current software solutions are fragmented across multiple platforms with varying degrees of usability,

documentation quality, and computational requirements. The TFM software ecosystem includes diverse implementations: ImageJ plugins such as Qingzong Tseng's PIV and FTTC implementations [17], JEasyTFM [18] and iTACS [19], the stand-alone tool Cellogram [20] for reference-free real-time analysis, MATLAB tools such as TFMLAB [21] for 4D TFM capabilities, the TFM toolkit by Huang et al., featuring Bayesian parameter selection [22] (available at https://github.com/CellMicroMechanics) or $\mu$-inferforce [8], implementing both FTTC and BEM algorithms, and Python tools such as pyTFM [23] and our own previous tool batchTFM [24]. For MSM analysis, pyTFM and iTACS are, to our knowledge, the only freely available solutions.

However, several challenges remain for researchers seeking to implement these techniques. First, the fragmented software landscape requires users to switch between different platforms and programming environments for complete analysis workflows. Second, selecting appropriate algorithmic parameters (regularization values, mesh densities, filtering parameters) requires significant expertise and often lacks real-time visual feedback during optimization. Third, processing time-series datasets at scale demands robust batch processing capabilities that many existing tools lack. Finally, researchers without programming expertise face substantial barriers to entry, limiting the techniques' adoption despite their biological value.

To this end, we developed napariTFM, a comprehensive TFM/MSM napari plugin that addresses these critical gaps through an intuitive graphical user interface, immediate feedback on parameter selection effects, and powerful batch processing capabilities for high-throughput experiments. The plugin leverages the Python-based napari image viewer [25], which is emerging as a powerful platform for biological image analysis with active community development and an extensive plugin ecosystem for comprehensive microscopy workflows.

The plugin supports both single-frame and time-series analysis of 2D microscopy images. napariTFM implements state-of-the-art algorithms including TV-L1 optical flow for displacement analysis, FTTC for force reconstruction, and finite element methods for stress calculation.

## 2 Design and implementation

### 2.1 Assumptions and limitations

napariTFM addresses the methodological complexities of force microscopy by making established computational approaches accessible, transparent, and easy to use for biologists conducting diverse experiments. The plugin implements well-established algorithms including TV-L1 optical flow for displacement analysis [26], FTTC for force reconstruction [9,12], and finite element methods for stress calculation [6,23].

We chose these specific methods based on their favorable balance of computational efficiency, robustness to noise, and ability to handle the range of displacement magnitudes encountered in typical biological experiments. However, we acknowledge that alternative approaches may be preferable in specific contexts. These methods rely on fundamental assumptions about substrate and cellular material properties (linear elasticity, homogeneity), measurement conditions (2D imaging of inherently 3D systems), and force balance (which may not hold locally even when satisfied globally). Different algorithmic approaches handle these challenges differently, with trade-offs between computational efficiency, noise robustness, and accuracy under various experimental conditions [7,12].

Important considerations for users include the effects of spherical aberration at image edges in large monolayers, and the current implementation of boundary conditions for MSM which work only when cell borders are clearly visible. Future versions will address boundary conditions for confluent layers without visible borders and could incorporate multiple algorithmic options with guidance for users on method selection.

### 2.2 Software architecture and implementation

napariTFM is implemented as a Python 3.6+package designed to integrate with the napari image viewer as an optional plugin. The complete analysis workflow from raw input data through preprocessing, displacement analysis, force calculation, and stress analysis is outlined in Fig 1.

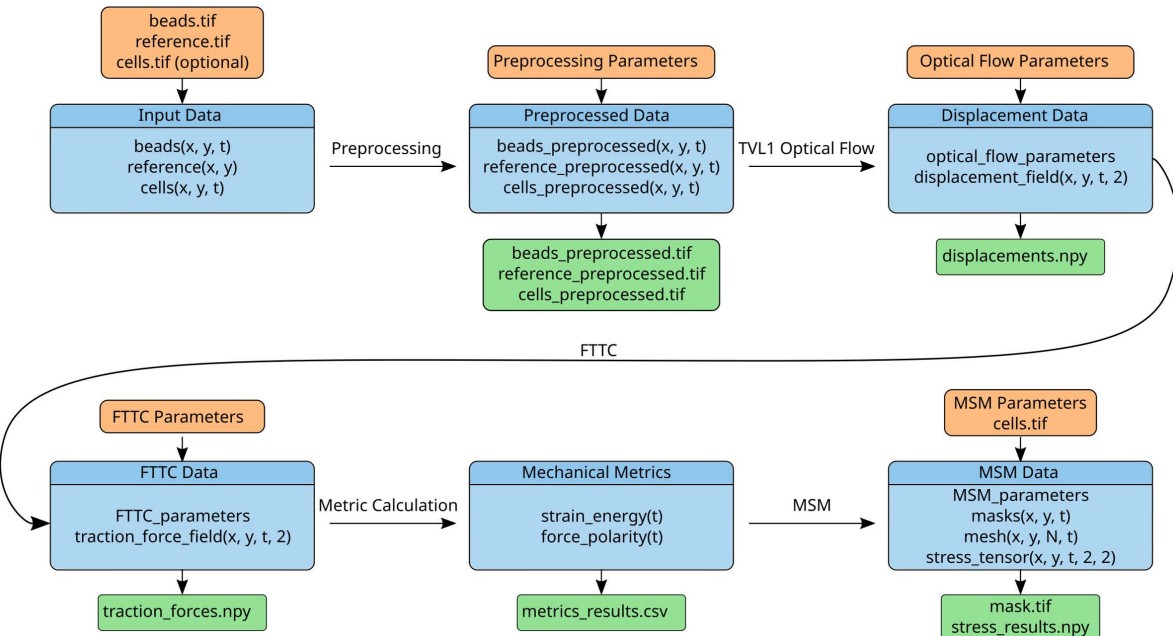

**Fig 1. napariTFM workflow and data structure.** Schematic overview of the napariTFM analysis pipeline showing the flow from raw input data (beads, reference, and optional cell images) through preprocessing, displacement analysis using TV-L1 optical flow, force calculation via FTTC (Fourier Transform Traction Cytometry), and stress analysis using MSM (Monolayer Stress Microscopy). Orange boxes indicate input data, blue boxes show analysis steps and their internal data structures, and green boxes indicate output files generated at each step.

The software provides both a graphical user interface through napari (Fig 2) and a standalone Python library for programmatic access. The core computational components utilize established open-source packages including NumPy for array operations, SciPy for scientific computing, OpenCV for image processing, scikit-image for automated detection and drift correction, and matplotlib for visualization.

The plugin organizes data using a hierarchical structure where each experimental field of view is represented as a frame containing multiple image layers. Input images (substrate in tensed and relaxed states, and optional cell images) are stored as separate layers, with analysis outputs added as additional layers during processing. Cell masks are created through simple threshold segmentation of cell images or can be provided externally as TIFF files.

## 2.3 Displacement field calculation

Optical flow algorithms have been successfully applied to TFM [26–28], where they have been shown to outperform classical PIV approaches in terms of both computational efficiency and accuracy [26,29]. napariTFM implements the TV-L1 variant [30,31], which is particularly well-suited for TFM analysis due to its ability to handle steep gradients in displacement fields, large displacements through multi-scale analysis, and provision of sub-pixel accuracy while being robust to intensity variations. The inherent smoothness constraint in TV-L1 optical flow makes it particularly robust to imperfect experimental conditions such as bead aggregates or non-uniform bead densities, reducing the need for post-processing steps like outlier removal that are sometimes necessary with other displacement tracking methods. However, users should be aware that this smoothness constraint can also over-regularize displacement fields in regions with genuinely high deformation gradients.

The TV-L1 algorithm minimizes an energy functional that combines brightness constancy assumptions (beads maintain intensity) with total variation regularization (smooth displacement fields) and additional constraints for numerical

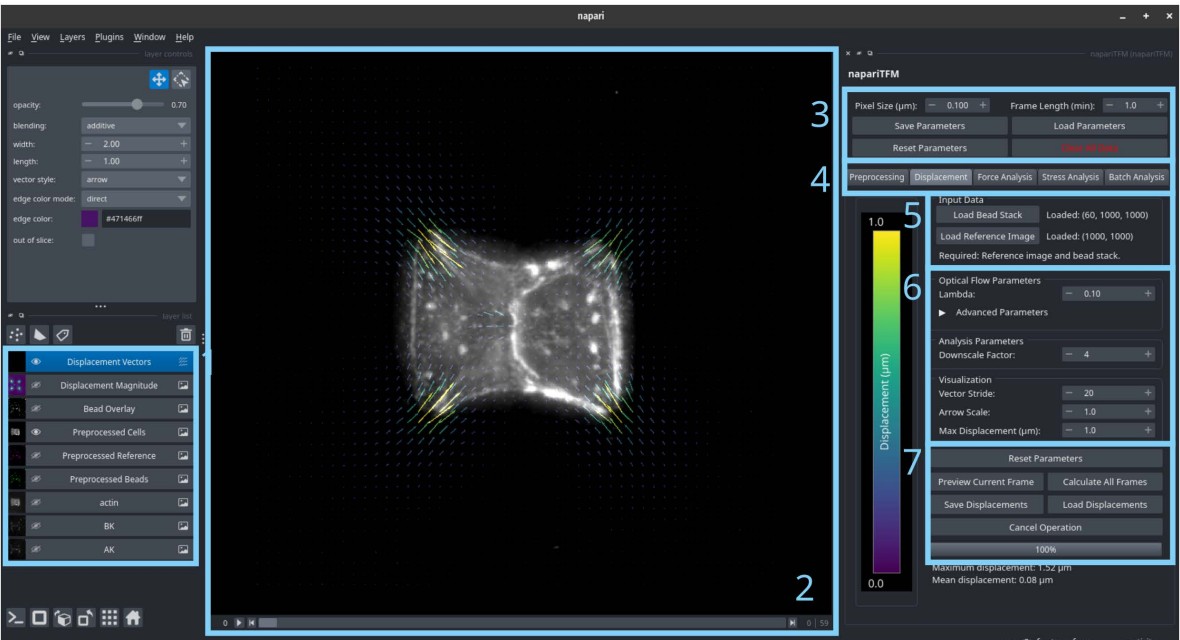

**Fig 2. napariTFM user interface in napari.** Screenshot of the napariTFM plugin integrated within the napari image viewer, showing: (1) the main image display with fluorescent bead data and overlaid displacement vectors, (2) napari's layer controls, (3) global parameters panel, (4) tabs that access controls for different analysis steps, (5) data input panel, (6) local parameters for the selected step, and (7) action controls (analyze, save data, preview etc.). The interface provides real-time parameter adjustment and visualization of results.

stability. It uses a multi-scale pyramid approach where images are analyzed at different resolution levels. Large displacements are captured at coarse scales while fine details are refined at higher resolutions. Key algorithm parameters include lambda ($\lambda$) which controls the balance between data fitting and smoothness (typical values 0.01–1.0, with lower values providing more smoothing for noisy data and higher values preserving detail for clear images), pyramid scales (number of resolution levels, typically 3–5), warps (number of iterative refinements per scale), epsilon (stopping criterion for optimization), and scale step (factor between pyramid levels, typically 0.5–0.8). Global image drift correction is performed using phase cross-correlation of the entire image pair, followed by image alignment and cropping to the overlapping field of view.

## 2.4 Traction force reconstruction

Traction forces are computed using Fourier Transform Traction Cytometry (FTTC) [9,10], which exploits the convolution theorem to efficiently relate substrate displacements to cellular tractions in Fourier space. FTTC typically processes images in seconds, making it well-suited for batch processing of large time-series datasets. napariTFM uses the Python implementation by Blumberg et al. [12].

For larger cell patches where substrate thickness becomes relevant, napariTFM also includes corrections for finite substrate thickness [23,32].

The reconstruction of traction forces from displacement data is an inverse problem and as such inherently ill-posed: small errors in displacement measurements can lead to large errors in the recovered forces [33]. Therefore, in order to avoid overfitting the traction force solution to noise, regularization is required. To this end, napariTFM uses Tikhonov regularization, which suppresses noise-dominated high-frequency contributions to the traction field [10,33].

The balance between fitting the displacement data and suppressing noise is controlled by the regularization parameter $\lambda$. napariTFM supports both automatic parameter selection via Generalized Cross-Validation (GCV) and manual specification of the regularization parameter.

When comparing datasets quantitatively, it is essential to maintain consistent regularization parameters to ensure that differences in recovered forces reflect biological variation rather than analysis artifacts. However, if substrate rigidity differs between conditions, the regularization parameter should be adjusted accordingly as the optimal regularization depends on the mechanical properties of the system.

A Lanczos filter with user-defined exponent is applied for additional noise reduction, with higher exponents providing stronger smoothing at the cost of spatial detail. The choice of Lanczos exponent, like the regularization parameter, should remain consistent within comparative studies.

## 2.5 Monolayer stress microscopy

Monolayer Stress Microscopy (MSM) calculates internal stress distributions within cell monolayers from measured traction forces [6,13]. The method models the cell monolayer as a thin elastic sheet and uses force balance to compute intercellular stresses from the traction field.

napariTFM implements MSM using a finite element method based on the pyTFM package [23], which uses SolidsPy [34] for solving the FEM equations. We extended this implementation by using Gmsh [35] for mesh generation, which provides improved mesh quality and user-controllable mesh parameters compared to the original pyTFM implementation.

Since TFM ensures global but not local force balance, unbalanced forces and torques are corrected before solving [13,23].

The stress calculation is largely independent of assumed material properties: it does not depend on Young's modulus and is only weakly influenced by Poisson's ratio [13].

Alternative approaches such as Bayesian inference have been developed that do not require assumptions about monolayer rheology [36], but are not currently implemented in napariTFM.

## 2.6 Synthetic data generation for validation

To validate the accuracy of napariTFM's displacement analysis, traction force reconstruction, and stress calculation algorithms, we generated synthetic datasets with known ground truth values for systematic comparison.

For traction force microscopy validation, we utilized the DirectMethod repository from Blumberg et al. [12], which provides a forward simulation approach to generate substrate displacements from known force fields. We programmed two force dipoles with identical centers and 90-degree rotation relative to each other, mimicking the characteristic force field pattern of a cell doublet on H-shaped micropatterns, similar to the experimental conditions described in our previous work [37].

The dipoles were applied on a circular region of radius 3 μm, and we varied the total force magnitude across three conditions (high: 0.25 μN, medium: 0.025 μN, low: 0.0025 μN) to span three orders of magnitude and probe the limits of displacement reconstruction. Substrate displacements were computed using the DirectMethod forward model assuming a linear elastic substrate with properties matching typical polyacrylamide hydrogels (Young's modulus $E = 20kPa$, Poisson's ratio $\nu = 0.5$).

To create realistic synthetic bead images, we used OpenCV to deform experimental fluorescent bead images according to the displacement maps generated by the forward simulation.

For monolayer stress microscopy validation, we employed two complementary approaches. First, we used an analytically solved problem consisting of a square plate under uniform loading, where both the internal stress distribution (constant within the plate) and boundary forces are known exactly, providing a rigorous benchmark for algorithm accuracy.

Second, we employed a finite element method (FEM), inspired by the modeling approach in [38,39] to generate realistic stress maps and corresponding traction force distributions that capture the mechanical behavior of migrating cells. This approach enables validation under more complex and biologically relevant conditions.

In this framework, cells are modeled as a two-dimensional active elastic solid via

$$\sigma_{ij} = \frac{h_c E_c}{1 + \nu_c} \left( \varepsilon_{ij} + \frac{\nu_c}{1 - \nu_c} \varepsilon_{kk} \delta_{ij} \right) + \frac{h_c E_c}{2(1 - \nu_c)} P_0 \delta_{ij} \, ,$$

(1)

where the first term represents the constitutive relation of a linear elastic material, with $\sigma_{ij}$ and $\varepsilon_{ij}$ denoting the Cauchy stress and strain tensor, respectively. The second term introduces an active contractile stress with constant contractility $P_0$. The cell layer is further characterized by the Young's modulus $E_c$, Poisson's ratio $\nu_c$, and effective contractile thickness $h_c$. To increase geometric and mechanical complexity, the cell is assumed to adhere at *19* distinct adhesive islands positioned near the periphery similar to [40]. Force balance is then given by

$$\partial_j \sigma_{ij} = Y_s(\mathbf{x}) u_i,$$

(2)

where $Y_s(\mathbf{x}) \neq 0$ at adhesion sites (and vanishes otherwise), describing the spring stiffness of the elastic substrate, and $\mathbf{u}$ is the substrate displacement field [41]. For the parametrization of this continuum model, we closely follow the parameters listed in the Supporting Information of [40]. In our FEM simulations, an initially round cell undergoes isotropic contraction until the force balance in Eq 2 is reached. The resulting cell shape and internal stress patterns are shown in Fig 4C (upper left row).

## 3 Results

We systematically validated napariTFM's performance across all major analysis components using synthetic datasets with known ground truth values, followed by demonstration of its capabilities on real experimental data. The validation encompassed displacement field calculation, traction force reconstruction, and monolayer stress microscopy analysis, confirming the software's accuracy and reliability for quantitative cellular force measurements.

### 3.1 Displacement analysis performance

The displacement field reconstruction achieved excellent accuracy across all tested scenarios (Fig 3A), with correlation coefficients between calculated and ground truth displacement of $\approx 0.93$ for low displacement scenarios and 0.99 for medium and high displacement conditions, demonstrating robust performance across biologically relevant deformation magnitudes.

Error analysis as a function of ground truth displacement magnitude (Fig 3B) revealed systematic performance characteristics across the measurement range. For displacements below approximately 1 pixel, sub-pixel accuracy limitations lead to systematic underestimation of displacement magnitudes. The optimal operating regime lies between roughly 1–10 pixels, where relative errors are minimized and remain close to zero. Above 10 pixels, the algorithm begins to show increasing variability, though correlation coefficients remain high even at 30 pixels of displacement.

For a 60x objective with a camera pixel size of 6.5 μm (corresponding to the imaging conditions used here), this optimal operating range translates to physical displacements of approximately 0.1–1 μm, with sub-pixel accuracy enabling detection of displacements as small as 0.01 μm, as demonstrated by the low displacement scenario. Depending on substrate stiffness (typically 1–50 kPa for polyacrylamide gels), this displacement range enables measurement of cellular traction forces from tens of Pa to several tens of kPa.

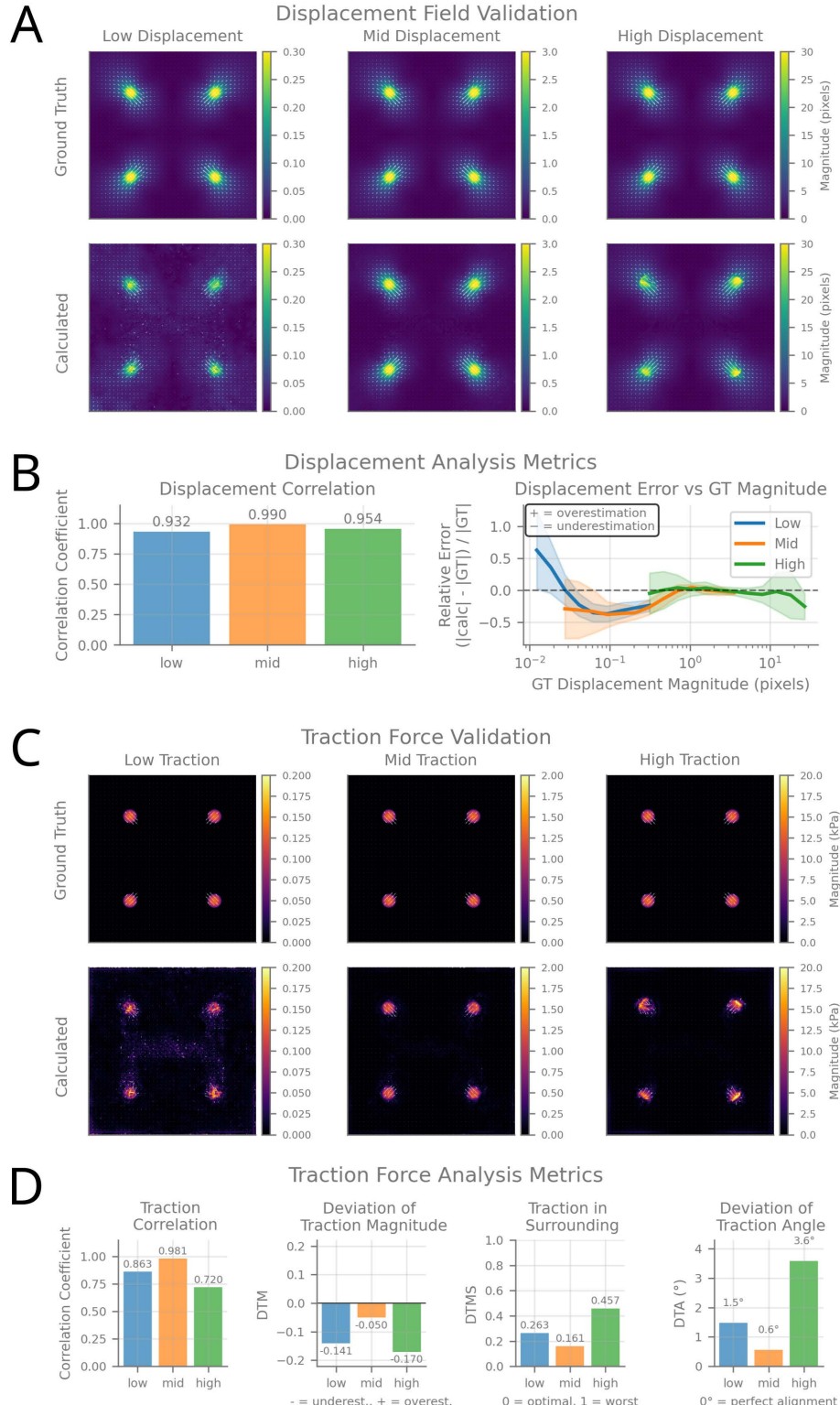

Fig 3. **Validation of displacement analysis and traction force calculation.** (A) Displacement field validation across three scenarios (low, mid, high displacement magnitudes). Top row shows ground truth displacement fields, bottom row shows calculated results using napariTFM's TV-L1 optical flow algorithm. Vector overlays indicate displacement direction and magnitude, with color maps showing displacement magnitude in pixels. **(B)** Displacement

analysis metrics. Left: correlation coefficients between calculated and ground truth displacement fields. Right: relative error as a function of ground truth displacement magnitude, showing systematic bias at different displacement scales. **(C)** Traction force validation using FTTC algorithm across three force magnitude scenarios. Top row shows ground truth traction fields, bottom row shows calculated results. Color maps indicate traction force magnitude in kPa, with vector overlays showing force direction. **(D)** Traction force analysis metrics. From left to right: correlation coefficients between calculated and ground truth traction forces, deviation of traction magnitude (DTM), traction in surrounding regions (DTPS) and deviation of traction angle (DTA).

## 3.2 Traction force reconstruction accuracy

Force reconstruction showed high fidelity to ground truth data (Fig 3C), with correlation coefficients ranging from 0.72 to 0.98 across different force magnitudes (Fig 3D). For further error quantification, we calculated the deviation in traction magnitude (DTM), the Traction in surrounding regions (DTMS) and the deviation of traction angle (DTA), as defined by Sabass et al. [10].

The medium force scenario performed best across all metrics, with minimal systematic bias (DTM $\approx$ −0.05), good spatial localization (DTPS = 0.16), and near-perfect directional accuracy (DTA = 1.5°).

The low force scenario showed reduced performance, likely due to increased noise, leading to underestimated forces (DTM = −0.141) and poor spatial localization (DTPS = 0.26).

The high force scenario suffered from large displacement artifacts, resulting in underestimated forces (DTM = −0.17), the highest spurious traction in surrounding regions (DTPS = 0.46), and reduced directional accuracy (DTA = 3.6°).

These findings indicate an optimal operating regime of approximately 1–10 pixels displacement, providing practical guidance for substrate stiffness selection in experimental design.

## 3.3 Monolayer stress microscopy performance

MSM validation using analytical square plate solutions (Fig 4A and 4B) yielded correlation coefficients above 0.98 for all stress tensor components ($\sigma_{xx}$, $\sigma_{yy}$, and their average $\sigma_{normal}$), with mean absolute and relative errors below 5%, indicating accurate recovery of both stress field structure and magnitude.

Validation using FEM simulations with realistic, cell-like geometries (Fig 4C and 4D) resulted in correlation coefficients ranging from 0.85 to 0.91. While relative errors were high for stress values near zero due to small normalization denominators, these errors decreased and transitioned to negative values at higher magnitudes. Ultimately, all substantial stress components were systematically underestimated by approximately −25%.

## 3.4 Validation against published experimental data

To further validate napariTFM's accuracy on real experimental data, we reanalyzed microscopy images from previously published experiments [37] and compared the results with the original analysis. The original analysis used a custom MATLAB-based pipeline that included Gaussian smoothing of displacement fields after single particle tracking analysis of bead movements, while napariTFM uses TV-L1 optical flow with an optional post-processing median filter which was not used here.

Qualitative comparison of displacement, traction force, and stress fields showed good agreement between methods (Fig 5A and 5B). The spatial patterns of force localization at cell-substrate adhesion sites and the overall stress distributions were highly consistent across individual cells and population averages, confirming that both analysis pipelines capture the same underlying mechanical behavior.

Quantitative analysis across 29 cells revealed close agreement between methods (Fig 5C). The median average displacement magnitudes differed by less than 0.1 μm, average traction forces by approximately 0 Pa, and average stress components by less than 1 mN/m.

Notably, napariTFM produced sharper displacement and traction force fields compared to the original analysis, as evident from the more localized force patterns and reduced spatial spreading in Fig 5A and 5B. This difference reflects the

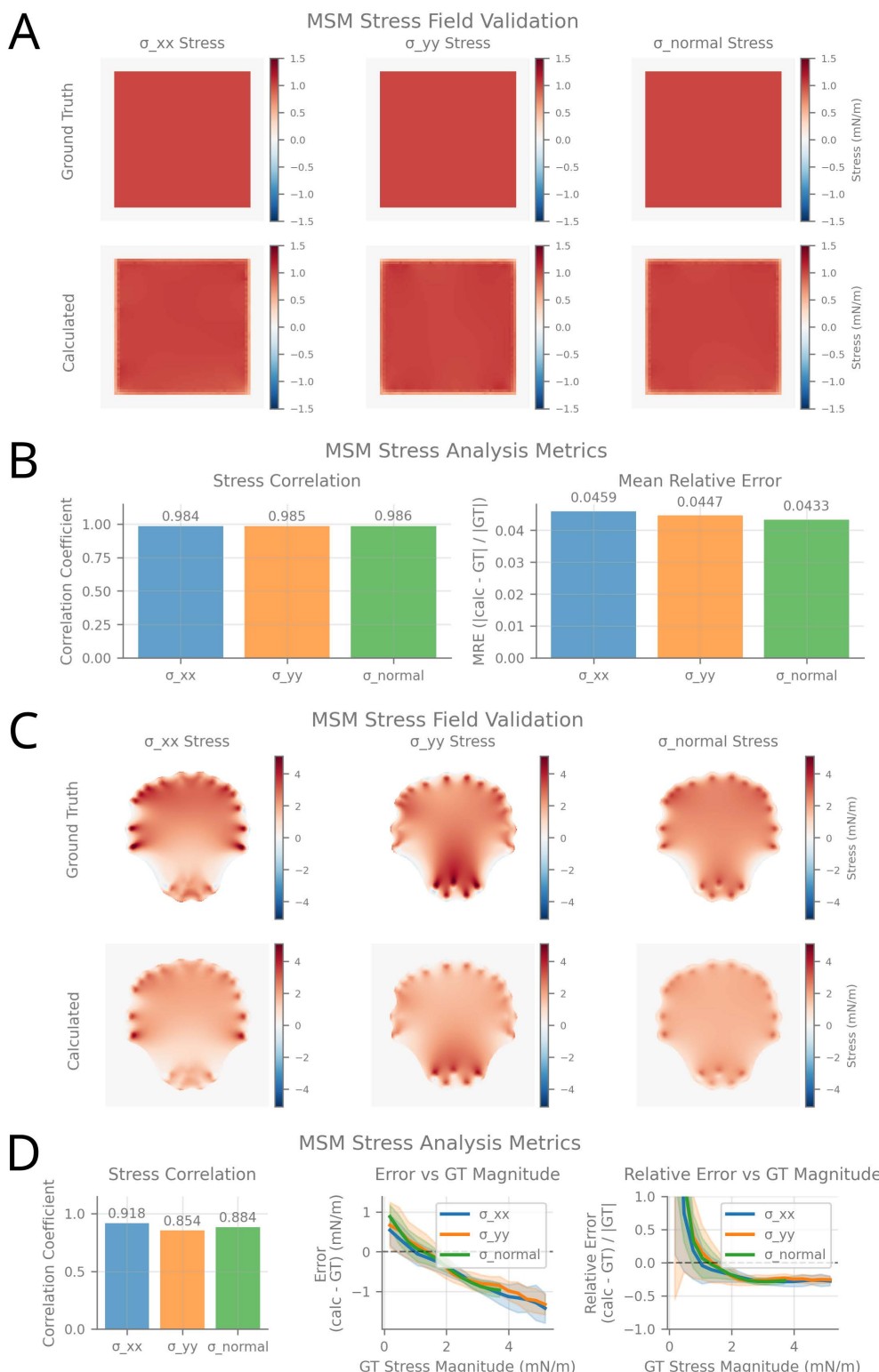

**Fig 4. Validation of Monolayer Stress Microscopy (MSM) analysis. (A)** Square plate analytical validation using a simple geometric test case with a known analytical solution. Stress tensor components ($\sigma_{xx}$, $\sigma_{yy}$, $\sigma_{normal}$) are shown for ground truth (top row) and MSM-calculated (bottom row) fields. **(B)** Square plate analysis metrics. Bar charts show correlation coefficients between calculated and ground truth stress fields for each stress component,

as well as mean relative error. **(C)** FEM-simulation–based validation using computational data from a finite element model resembling a snapshot of a migrating cell. Stress tensor components are shown for ground truth (top row) and MSM-calculated (bottom row) fields, with layout as in **(A)**. **(D)** FEM-simulation validation metrics. Correlation coefficients between calculated and ground truth stress fields are shown for each stress component, as well as absolute and relative error with respect to ground truth sress magnitude.

absence of Gaussian smoothing in napariTFM's workflow. As a consequence, maximum displacement and traction force values were consistently higher with napariTFM, while spatially-averaged quantities remained comparable between methods. The sharper fields obtained with napariTFM may provide improved spatial resolution for identifying localized mechanical features such as focal adhesion positions or stress concentrations.

## 4 Availability and future directions

napariTFM addresses a critical need in the cellular mechanics community by providing a comprehensive, user-friendly platform that integrates state-of-the-art algorithms for both traction force microscopy and monolayer stress microscopy. Our validation results demonstrate that the software achieves excellent accuracy across the full range of biologically relevant force and displacement magnitudes, with correlation coefficients consistently above 0.9 for most scenarios.

The integration within the napari ecosystem represents a significant advantage for the broader microscopy community, leveraging napari's established plugin architecture and intuitive interface design. The real-time parameter adjustment and immediate visualization capabilities address a common limitation of batch-processing tools, allowing users to optimize analysis parameters interactively and assess data quality during processing.

Our synthetic data validation revealed important performance characteristics across different experimental conditions. The reduction in accuracy for low displacement scenarios reflects the fundamental signal-to-noise limitations inherent to TFM analysis. This finding is consistent with previous studies and highlights the importance of experimental design considerations, such as substrate stiffness selection and imaging quality optimization, for achieving optimal force measurement accuracy.

While napariTFM's primary contribution lies in integrating established methods into an accessible platform, it is important to acknowledge the algorithmic choices and their limitations. Our validation demonstrates the accuracy of specific implementations: TV-L1 optical flow, FTTC, and finite element MSM. We selected these methods for their favorable balance of computational efficiency, noise robustness, and ability to handle typical biological displacement magnitudes, but alternative approaches (e.g., PIV, BEM) may offer advantages in specific contexts [7,8]. Future versions could incorporate multiple algorithmic options with guidance for method selection based on experimental requirements.

Specifically, Particle Image Velocimetry (PIV) and single particle tracking (SPT) methods have been widely used in TFM and are still the norm up to date. Digital Image Correlation (DIC), including its 3D variant Digital Volume Correlation (DVC) [42,43], provides robust displacement tracking for volumetric datasets and has been successfully applied to 3D TFM. While Python implementations of DIC exist (e.g., spam [44]), the ecosystem for these methods is less mature than for optical flow. Our current implementation uses OpenCV's C++-based TV-L1 implementation for computational efficiency, but we envision future versions that could leverage napari's extensibility to offer users a choice of displacement detection algorithms. Such integration could provide a unified interface for comparing different approaches on the same dataset.

All force microscopy methods rely on assumptions that users must understand when interpreting results. TFM assumes linear elastic substrate behavior to relate measured displacements to cellular traction forces, which means that substrates with more complex material properties, such as matrigel or collagen gels, require different methods. For MSM, while the formulation invokes elastic sheet theory, the calculated stress distributions are largely independent of the assumed elastic modulus and only negligibly influenced by Poisson's ratio [13], making the elasticity assumption less restrictive than it may appear. Standard 2D analysis captures projections of 3D mechanical systems, potentially missing out-of-plane forces. Regularization in FTTC represents critical trade-offs between noise suppression and spatial detail preservation;

 

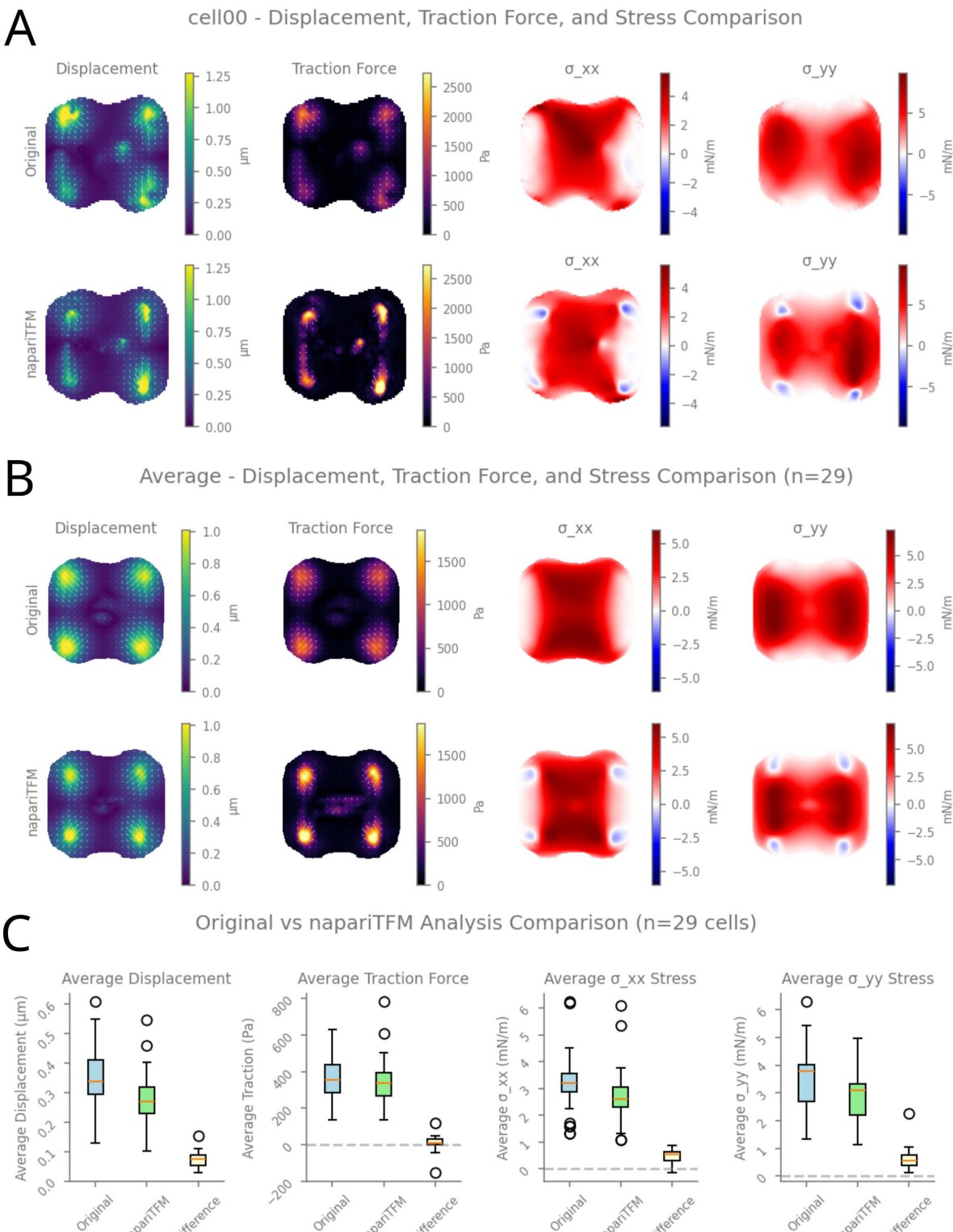

Fig 5. **Validation of napariTFM against experimental data, previously published in eLife [37]. (A)** Single-cell analysis showing displacement fields (left column), traction force fields (second column), and stress tensor components $\sigma_{xx}$ (third column) and $\sigma_{yy}$ (right column). Top row: original analysis from Ruppel et al., eLife 2023. Bottom row: reanalysis using napariTFM. Color maps indicate displacement magnitude in μm, traction force magnitude

in Pa, and stress components in mN/m. **(B)** Population-averaged fields across 29 cells. Top row: averages from original analysis. Bottom row: averages from napariTFM reanalysis. Layout as in **(A)**. **(C)** Quantitative comparison of analysis outputs. Box plots show distributions of average displacement magnitude (left), average traction force (second from left), average $\sigma_{xx}$ stress (second from right), and average $\sigma_{yy}$ stress (right). For each metric, three conditions are shown: Original (original published analysis), napariTFM (reanalysis), and Difference. Horizontal dashed line at zero indicates no difference between methods.

napariTFM's real-time visualization allows interactive assessment, but optimal parameters ultimately require biological judgment about expected force patterns. MSM requires force balance corrections because TFM ensures global but not local equilibrium. The magnitude of these corrections can serve as a quality metric for the analysis.

Important practical considerations include distinguishing between isolated cells and confluent monolayers, which require different analytical approaches and boundary conditions. Our current MSM implementation works only when cell borders are clearly defined. Boundary conditions for truly confluent tissues without visible cell-cell boundaries will be addressed in future versions. Additionally, spherical aberration at image edges and photobleaching in time-lapse experiments can affect displacement accuracy, emphasizing that careful experimental design remains essential for high-quality force microscopy regardless of computational methods.

Current limitations include the restriction to 2D analysis and the computational requirements for large-scale time-series datasets. Future developments will focus on extending the framework to 2.5D and 3D traction force microscopy, implementing GPU acceleration for improved processing speed, and developing specialized tools for high-throughput screening applications.

napariTFM fills an important gap in the TFM and MSM software ecosystem by combining an interactive and user-friendly graphical interface with state-of-the-art algorithms. Its open-source nature and integration within the napari platform position it as a valuable resource for advancing quantitative studies of cellular force generation and transmission across diverse biological systems.

napariTFM is open-source software available at https://github.com/ArturRuppel/napariTFM (GNU General Public License) and can be installed by following the installation procedure outlined in the readme. Synthetic validation datasets and analysis scripts are included in the repository. The dataset used in Fig 5 is available at https://zenodo.org/records/18390989.

## Acknowledgments

We acknowledge the foundational algorithmic contributions that enabled this work: Andreas Bauer and Ben Fabry for the pyTFM implementation of monolayer stress microscopy and finite substrate thickness corrections, and Johannes Blumberg and Ulrich Schwarz for the FTTC implementation and synthetic data generation methods. napariTFM builds upon these established frameworks to provide an integrated analysis platform for the cellular mechanics community. We further would like to thank François Graner for providing valuable feedback on the manuscript. During the development of this work, we used Claude (Anthropic) as a coding assistant for software development and to improve the clarity and organization of the manuscript text. We reviewed and edited all AI-generated content and code, and take full responsibility for the software implementation and final manuscript.

## Author contributions

**Conceptualization:** Artur Ruppel.

**Data curation:** Artur Ruppel.

**Formal analysis:** Artur Ruppel.

**Funding acquisition:** Martial Balland, François Fagotto.

**Investigation:** Artur Ruppel.

**Methodology:** Artur Ruppel.

**Project administration:** Artur Ruppel.

**Resources:** Artur Ruppel.

**Software:** Artur Ruppel.

**Supervision:** François Fagotto.

**Validation:** Artur Ruppel, Dennis Woerthmueller.

**Visualization:** Artur Ruppel.

**Writing – original draft:** Artur Ruppel.

**Writing – review & editing:** Artur Ruppel, Dennis Woerthmueller, Martial Balland, François Fagotto.

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
