## [Decision Letter · Decision Letter 0]

2 Dec 2025

napariTFM: An Open-Source Tool for Traction Force Microscopy and Monolayer Stress Microscopy

PLOS Computational Biology

Dear Dr. Ruppel,

Thank you for submitting your manuscript to PLOS Computational Biology. After careful consideration, we feel that it has merit but does not fully meet PLOS Computational Biology's publication criteria as it currently stands. Therefore, we invite you to submit a revised version of the manuscript that addresses the points raised during the review process.  While we hope you can address all the concerns of the reviewers, in particular, please focus on the concerns of Reviewer 1 with regards to validating the software on experimental data.  If accepted, after revisions, we also feel that this article would be better placed within the Software section of PLoS Computational Biology.

We look forward to receiving your revised manuscript.

Kind regards,

Joshua N. Milstein

Academic Editor

PLOS Computational Biology

Dimitrios Vavylonis

Section Editor

PLOS Computational Biology

**Journal Requirements:**

Potential Copyright Issues:

- The following Figure contains screenshots: Figure 2.. We are not permitted to publish these under our CC-BY 4.0 license, websites are usually intellectual property and are copyrighted.This includes peripheral graphics of the web browser such as icons and button. We ask that you please remove or replace it.

5) Please ensure that the funders and grant numbers match between the Financial Disclosure field and the Funding Information tab in your submission form. Note that the funders must be provided in the same order in both places as well.

**Reviewers' comments:**

Reviewer's Responses to Questions

**Comments to the Authors:**

Reviewer #1: The authors present a user-friendly graphical interface for estimating cellular traction forces from TFM images and describe its validation. The motivation is sound, as existing traction force algorithms are technically demanding and often inaccessible to experimental biologists. The proposed tool may indeed lower this barrier and contribute to wider adoption of TFM and MSM analyses.

However, the manuscript does not meet the PLOS Computational Biology criteria of “providing new biological insights” or “bringing exceptional new capabilities.” The presented work primarily integrates existing algorithms within a GUI rather than introducing novel computational approaches or enabling new biological discoveries.

The Methods section includes detailed mathematical formulations, but all algorithms are established methods from previous studies. As none are original, this section could be condensed with appropriate citations instead of full equation listings, which currently risk confusing readers.

In Figure 3, the validation using three displacement scenarios (low, medium, high) lacks justification, and the quantitative differences between these conditions are unclear. The use of correlation coefficients as the sole evaluation metric is problematic, as it measures only qualitative similarity. Quantitative error measures such as MSE or RMSE should be provided, and the method of calculating strain energy should be clarified.

Figure 4A shows nearly uniform stress distributions, raising doubts about the relevance of the test case. Similar concerns apply to the reliance on correlation coefficients in Figure 4. While the moving-cell model in Figures 4C–D is a positive addition, it still demonstrates software validity rather than “exceptional capability”. Figure 5 is described as showing spatial propagation of contractile forces, but the figure and description do not clearly support this claim.

As a minor issue, the reference list does not follow journal format: only the first six authors should be listed, followed by “et al.” Reference 22 and several others should be corrected accordingly.

In summary, while the software may have practical value, the manuscript lacks computational or biological innovation required for publication.

Reviewer #2: Traction force microscopy (TFM) and monolayer stress microscopy (MSM), which builds on it, have become standard tools in the field of mechanobiology, but as the authors correctly point out, what is missing is an accessible and GUI-based tool to run such reconstructions without having to worry much about the underlying methods and limitations. The new software napariTFM presented here fills this gap and could become a very useful resource for a large user community. The software puts together the most advanced solutions in this field and is available on GitHub. I did not have any trouble to install it. In the manuscript, the authors verify their software with simulated data and using experimental data from their own recent paper in eLife (Ruppel, Artur, et al. "Force propagation between epithelial cells depends on active coupling and mechano-structural polarization." Elife 12 (2023): e83588). In my assessment, this could be a very useful methods paper for a broad user community.

I have a few comments that should be addressed in a moderate revision.

First I note that the authors provide the software, but not the image data. All data files described in the manuscript should be made available in the GitHub repository and included in the ZIP-file. I particular, I note that Fig. 1 gives specific file names like beads.tif, so it would be nice to be able to try the software out with such files.

Regarding the image processing part, I wonder if it would be possible to add other methods than the optical flow routine. For example, in TFM, often one also uses Digital Volume Correlation (compare e.g. the work of Christian Franck). Naparia seems to be a good environment to provide other options for image processing; maybe some of these are already available through other plugins.

In the introduction, the authors should also mention the software provided by Benedikt Sabass on GitHub, https://github.com/CellMicroMechanics, especially for BayesianTFM (although it requires the commerical software Matlab, this code seems to be used widely).

Also in the introduction, BEM should be cited before FTTC and with Dembo BPJ 1999, because this was the very beginning of TFM. For FTTC, one should also cite Sabass BPJ 2008. For FEM, a pioneering paper was Yang, Z., Lin, J. S., Chen, J., & Wang, J. H. (2006). Determining substrate displacement and cell traction fields—a new approach. Journal of theoretical biology, 242(3), 607-616. In general, the manuscript would benefit from more references throughout.

References are not always complete. For example:

Ref. 15: book title missing

Ref. 22: journal title etc missing

The concept and need for regularization should be explained before methods for it are introduced.

I am a bit unsure if "research paper" is the right category for this work. PLOS CB also has the categories "Methods" and "Software". Maybe the latter would fit better.

**Have the authors made all data and (if applicable) computational code underlying the findings in their manuscript fully available?**

Reviewer #1: Yes

Reviewer #2: **No:** The software is on GitHub, but not the imaging data.

PLOS authors have the option to publish the peer review history of their article (what does this mean? ). If published, this will include your full peer review and any attached files.

**Do you want your identity to be public for this peer review?** For information about this choice, including consent withdrawal, please see our Privacy Policy .

Reviewer #1: No

Reviewer #2: **Yes:** Ulrich Schwarz

**Figure resubmission:**

**Reproducibility:**



---

## [Editor Report · Decision Letter 1]

18 Feb 2026

Dear %TITLE% Ruppel,

We are pleased to inform you that your manuscript 'napariTFM: An Open-Source Tool for Traction Force Microscopy and Monolayer Stress Microscopy' has been provisionally accepted for publication in PLOS Computational Biology.  Note, this manuscript will be published within the Software section.  As per policy on software submissions, the software must be downloadable anonymously in source code form and licensed under an Open Source Initiative (OSI) compliant license. The source code must be accompanied by documentation on building and installing the software from source, as well as for using the software, including instructions on how a user can test the software on supplied test data.

Best regards,

Joshua N. Milstein

Academic Editor

PLOS Computational Biology

Dimitrios Vavylonis

Section Editor

PLOS Computational Biology

---

## [Editor Report · Acceptance letter]

PCOMPBIOL-D-25-02104R1

napariTFM: An Open-Source Tool for Traction Force Microscopy and Monolayer Stress Microscopy

Dear Dr Ruppel,

I am pleased to inform you that your manuscript has been formally accepted for publication in PLOS Computational Biology. Your manuscript is now with our production department and you will be notified of the publication date in due course.

With kind regards,

Anita Estes
